# SOXC Transcription Factors as Diagnostic Biomarkers and Therapeutic Targets for Arthritis

**DOI:** 10.3390/ijms24044215

**Published:** 2023-02-20

**Authors:** Emad A. Ahmed, Abdullah M. Alzahrani

**Affiliations:** 1Biological Sciences Department, College of Science, King Faisal University, Al Ahsa 31982, Saudi Arabia; 2Lab of Molecular Physiology, Zoology Department, Faculty of Science, Assiut University, Assiut 71515, Egypt

**Keywords:** SOX4, SOX11, therapeutic targets, rheumatoid arthritis, osteoarthritis, inflammation

## Abstract

Osteoarthritis (OA) and rheumatoid arthritis (RA) are two common disorders that disrupt the quality of life of millions of people. These two chronic diseases cause damage to the joint cartilage and surrounding tissues of more than 220 million people worldwide. Sex-determining region Y-related (SRY) high-mobility group (HMG) box C, SOXC, is a superfamily of transcription factors that have been recently shown to be involved in various physiological and pathological processes. These include embryonic development, cell differentiation, fate determination, and autoimmune diseases, as well as carcinogenesis and tumor progression. The SOXC superfamily includes SOX4, SOX11, and SOX12, all have a similar DNA-binding domain, i.e., HMG. Herein, we summarize the current knowledge about the role of SOXC transcription factors during arthritis progression and their potential utilization as diagnostic biomarkers and therapeutic targets. The involved mechanistic processes and signaling molecules are discussed. SOX12 appears to have no role in arthritis, however SOX11 is dysregulated and promotes arthritic progression according to some studies but supports joint maintenance and protects cartilage and bone cells according to others. On the other hand, SOX4 upregulation during OA and RA was documented in almost all studies including preclinical and clinical models. Molecular details have indicated that SOX4 can autoregulate its own expression besides regulating the expression of SOX11, a characteristic associated with the transcription factors that protects their abundance and activity. From analyzing the currently available data, SOX4 seems to be a potential diagnostic biomarker and therapeutic target of arthritis.

## 1. Introduction

Osteoarthritis (OA) and rheumatoid arthritis (RA) are two forms of arthritis with a similar clinical phonotype, causing joint pain, stiffness, and probably disability, but occurring for different reasons and differing in pathophysiological mechanisms and in treatment strategies. These two common disorders affect more than 220 million people in the world. Osteoarthritis, the most common type, is a degenerative joint disease, characterized by the progressive deterioration of the articular cartilage or of the entire joint (the articular cartilage and the joint lining), the ligaments, and the subchondral bone. In addition, although inflammatory conditions can be associated with OA, basically, it is not an inflammatory disease. On the other hand, RA is a chronic systemic autoimmune inflammatory disorder characterized by the infiltration of inflammatory cells at the synovial lining resulting in hyperplasia and the destruction of cartilage and bone tissues (reviewed by Pap and Korb-Pap [1]). Typically, RA starts in the peripheral joints and progresses to damage the proximal joints if left without treatment. In the meantime, prolonged joint inflammation can destroy the joints and cause bone erosion and a loss of cartilage.

To date, there is no cure for OA or RA, and the available therapeutic protocols are able to partially reduce the pain, relieve the symptoms, improve quality of live and increase survival. However, an understanding of the involved signaling biomolecules and their targets and inhibitors is advantageous in designing new therapeutic arthritis drug or drug combinations. A little over 30 years ago, the gene of the male sex-determining region of the Y chromosome (SRY) was discovered, which later facilitated the discovery of the whole family of essential regulatory gene-encoding transcription factors (TFs) that control cell fate in many processes [2,3]. To date, 20 SRY-like box (*SOX*) genes have been reported in the mammalian genomes, which encode eight groups of SOX TFs, named from A to H. However, during the past decade, studies have found links between the dysregulation of sex-determining region Y-related (SRY) high-mobility group (HMG) box C (SOXC) levels and the progression of arthritis disorders. SOXC is a superfamily of TFs that have been recently shown by several studies to be involved in the onset and pathogenesis of arthritis. Interestingly, the great majority of these studies have been conducted during the last 5 years [4,5,6,7,8,9,10,11,12,13,14,15,16,17]. The *SOXC* gene family encodes SOX4, SOX11, and SOX12 TFs that contain a conserved high mobility group (HMG)-box domain by which SOXC proteins are able to bind to minor grooves in the DNA. SOXC TF binding to DNA can lead to alterations in the chromatin architecture associated with changes in the downstream genes at both transcriptional and functional levels. At least in vitro, while the three have similar DNA-binding abilities, the mammalian SOXC unique features in the SOX11 DNA binding domain allow the more potent transactivation function of SOX11; however, SOX12 was found to be the weakest [18].

In the current review, we aim to highlight SOX4 and SOX11 transcription factors as a diagnostic biomarker and therapeutic targets of arthritis diseases, and to clarify their role in regulating osteoarthritis and rheumatoid arthritis in relation to the involved downstream and upstream signaling molecules. An overview of the role of SOXC transcription factors in OA and RA has not been conducted before.

## 2. SOXC Transcription Factors under In Vivo Inflammatory Conditions Associated with Arthritis

At local inflammatory sites in humans, a specific type of CD4+ T cell was found to enhance ectopic lymphoid-like structure (ELS) formation through the production of the chemokine CXCL13 and other mediators [17]. These ectopic structures were proved to play crucial roles in the body’s response to infections and during cancer and autoimmune diseases [19]. For instance, ELSs’ immune activity was reported to be correlated with a better prognosis of cancer [20] and to stimulate the autoimmune response, i.e., via autoantibody production [21,22]. The function of the ectopic ELSs was reported to be maintained by fibroblast-like synoviocytes (FLSs), nonimmune cells located at the synovial tissues and strongly participating in the pathogenesis of RA. Related to that, in synovial CD4+ T cells from patients with RA, SOX4 was found to be significantly upregulated when compared with patients’ blood CD4+ T cells, which was further correlated with CXCL13 production and ELS formation in RA synovium [17]. Similarly, the *SOXC* genes, *SOX4* and *SOX11* are involved in synovial hyperplasia during arthritic disease through enhancing the survival of FLS, but do not affect the FLS proliferation rate [23]. Further mechanistic analysis has revealed that SOXC proteins directly upregulate the expression of the tumor necrosis factor (TNF-α) target genes, which support FLS transformation in arthritic disease [23]. Moreover, besides promoting FLS survival, SOXC genes, especially SOX4, were found to enhance the FLS invasion of joint cavities and their migration over plastic surfaces. This function of SOX4 during arthritis seems to be similar to its role in cancer, where it promotes metastasis and epithelial mesenchymal transition [24]. In the meantime, the downregulation of *SOXC* genes expression in the articular chondrocytes synovial lining significantly decreased synovitis and the erosion of the articular cartilage [23]. Thus, these specialized cells inside the joint synovium that are crucial in joint inflammation and destruction are regulated by the SOXC TFs.

In healthy tissues, FLSs populate at the synovial lining of the joints and maintain the synovial fluid homeostasis through producing cartilage-protecting proteins and enhancing joint lubrication. However, in arthritic inflammatory conditions, FLSs are epigenetically changed and transformed into a main source of catabolic enzymes and inflammatory cytokines that promote the degeneration of joints [25]. In addition, the formed ectopic structures, ELSs, acquire cancer-like characteristics and contribute aggressively to joint degeneration [17]. In this regard, in patients with OA, SOX4 is elevated in the inflamed synovium-containing FLS relative to the non-inflamed synovium [23]. Consistent with that, SOX4 was shown to be regulated by the ROS/TGF-β signal and to participate in the pathogenesis of OA via increasing FLS senescence [5]. Among the signaling pathways that regulate FLS transformation, the NF-κB signaling transcription factor RELA/p65, the downstream of TNF-α, was shown to play a crucial role in joint degeneration, synovial hyperplasia, and bone loss [26]. Related to that, in a genome-wide association study investigating the transcriptional activities and DNA binding of SOX4 and RELA/p65, SOX4 and RELA were found to physically interact with each other on the chromatin and orchestrate several processes to regulate the expression of the downstream genes of TNF and then enhance the transformation of FLSs in arthritis [10]. Together these data reflect the pivotal role of SOXC TFs, especially SOX4, in regulating FLS and ELS actions under the in vivo inflammatory conditions associated with arthritis.

## 3. SOXC Transcription Factors as Potential Diagnostic Biomarkers of Arthritis

Developmental biology studies support the idea that SOX4 and SOX11 are essential in promoting articular and bone formation [13,18]. However, according to both the preclinical and the clinical studies, the progression of arthritis diseases was found to be associated with dysregulated levels of SOX4 and SOX11 TFs (summarized in Table 1 and Table 2), which indicates that both could be used as diagnostic biomarkers of arthritis; however, there is still little data available regarding the role of SOX11 in RA. The upstream signaling molecules are advanced glycation end products (AGEs), tumor necrosis factor alpha (TNF), secreted modular calcium-binding protein 2 (SMOC2), tumor necrosis factor (TNF-β), the long non coding (Lnc-PATR1), and the miRNAs miR-31-5p and miR-373-3p. However, the downstream target molecules include disintegrin-like and metallopeptidase with thrombospondin types 4 and 5 motif (ADAMTS4 and 5, respectively), the transcription factor RELA, the motor protein myosin1c (MYO1C), the protein kinase (ERK), the mechanical target of rapamycin kinase (mTORC1), Lnc MCM3AP-AS1, and miR149-5p.

### 3.1. SOX4 as a Potential Diagnostic Biomarker of Osteoarthritis

The progression of OA disorder was recently documented in several studies to be associated with upregulated levels of SOX4 (summarized in Table 1). In this regard, higher levels of SOX4 were seen in both OA models of chondrocytes and in OA patients [7]. In addition, SOX4 was elevated in inflamed arthritis patients’ synovium compared to non-inflamed synovium and then was considered to be an early diagnostic biomarker during OA pathogenesis [5]. Consistent with that, treating chondrocytes with the inflammatory agent AGEs upregulated the level of SOX4 in a dose-dependent manner in vitro [6]. Similarly, SOX4 is upregulated in OA patients and IL-1β-treated chondrocytes [9]. Moreover, SOX4 was suggested to be involved in osteoarthritis onset, where its mRNA expression increased in the cho cartilage of patients with osteoarthritis compared to the control subjects [8]. In all, SOX4 was shown to be associated with the onset and progression of OA, suggesting it as a potential diagnostic biomarker of OA.

### 3.2. SOX4 as a Potential Diagnostic Biomarker of Rheumatoid Arthritis

As indicated above, RA is basically chronic inflammatory disease, in which a high level of SOX4 seems to be required for the progression of inflammatory conditions at a local site. This has been evidenced by the fact that SOX4 is upregulated in both RA patients and RA mouse models [10,11,17,27,28]. The mechanistic actions associated with SOX4 upregulation and regulatory function during RA are discussed in detail below. However, as shown in Table 1, based on the data, SOX4 appears to be a potential biomarker for reflecting the progression of RA diseases.

### 3.3. SOX11 Is Dysregulated during Osteoarthritis

Unlike SOX4, which is upregulated in almost all of the studies investigating its role in arthritis disease, SOX11 is downregulated in some studies and upregulated in others during osteoarthritis progression. For instance, SOX11 was reported to be upregulated in both patients with OA and IL-1β-treated chondrocytes, which was associated with osteoarthritis progression via the induction of TNF-α [12]. Similarly, *SOX11* gene expression was upregulated in the knee cartilage of OA patients [5]. In synchronization with this, SOX11 inhibition decreased the level of TNF-α in patients with OA and in chondrocytes treated with IL-1β in vitro [12]. In addition, SOX4 and SOX11 upregulation enhanced articular cartilage degeneration in cell lines and OA patient samples [8]. However, based on reverse-phase protein arrays conducted on the tissues of OA patients and normal chondrocytes, SOX11 was identified among the novel proteins that are downregulated during arthritis [30]. In addition, SOX11 protein was shown to be expressed in the articular cartilage of healthy mature mice, but its level decreased in the cartilage of osteoarthritis mouse models [13]. In agreement with this, in human chondrocyte cells and in murine OA models, SOX11 downregulation was reported to be associated with inflammatory symptoms [14]. On the other hand, developmental biology data has indicated that SOX11 is a critical regulator of chondrogenesis where SOX11-modified rat mesenchymal stem cells (rMSCs) were found to have clinical implications in accelerating the healing of cartilage defects [29]. Moreover, the deletion of SOX11 from mice was lethal at birth where SOX11-deficient embryos displayed skeletal malformations indicating the importance of SOX11 for articular and bone formation [31]. Altogether, studies dedicated to exploring the role of SOX11 in arthritis diseases have shown that it is dysregulated during arthritis diseases progression, but more mechanistic details are still needed. However, SOX11 appears to be crucial for chondrogenesis and cartilage regeneration.

## 4. Signaling Mechanisms Involved in SOXC TFs Promoting Arthritis

Recognizing more molecular details about OA and RA and the involved signaling molecules and factors is important and very useful for designing effective therapies. As shown above, SOX11 is elevated in patients with OA, indicating its involvement in arthritis progression. In this way, SOX11 has been found to promote OA through the activation of TNF, where SOX11 inhibition reduced the molecular biomarkers of OA and apoptosis, as well as enhanced OA cell proliferation, suggesting that SOX11 could improve the proliferation of cartilage cells [12,16]. However, according to other studies, SOX11 is downregulated during arthritis, suggesting that it is required to prevent arthritis; therefore, further molecular details are still needed to clarify the role of SOX11 in arthritis [13,14,30].

In cancer, we and others have recently clarified that SOX4 is a master regulator of the epithelial-mesenchymal transition, an important pathological process associated with cancer and chronic inflammation which enhances the propagation of epithelial cells [11,23]. In a similar mechanism, SOX4 was recently shown to be involved in promoting arthritis and in regulating FLS activities under both normal and inflammatory conditions.

Mechanistically, SOX4 promoting arthritis has been documented to occur through regulating several signaling pathways. (1) As a critical mediator of the TNF-induced transformation of FLS, SOX4 interacts with RELA (NF-κB signaling molecule) to regulate the expression of TNF downstream genes and thus maintains the transformation of FLS and inflammatory pathology in arthritis [10,23]. (2) SOX4 is regulated by the ROS/TGF-β signal to enhance OA pathogenesis and FLS senescence [5]. (3) SOX4 is involved in osteoarthritis onset by increasing the levels of two major aggrecanase-degrading articular cartilage enzymes, Adamts4 and Adamts5, through binding to their gene promoters [8]. The degradation of the cartilage extracellular matrix (ECM) is one of the main features associated with arthritis. The degradation of aggrecans in the ECM of aggrecan was found to regulated by these two enzymes. Thus, SOX4 appears to regulate the level of these enzymes and then promote arthritis. (4) SOX4 is upregulated in synovial CD4+ T cells and contributes to the production of CXCL13 and the formation of ELSs at the inflammatory sites in RA patients [17]. (5) SOX4 activated the long noncoding MCM3AP-AS1, aggravated OA progression via targeting the miR-149-5p/Notch1 axis and then modulated autophagy and ECM degradation [7].

## 5. SOX4 as a Therapeutic Target of Arthritis

Targeting the SOXC proteins or the mechanisms that stabilize them during arthritis through the systemic delivery of inhibitors has been suggested as an option to be explored [28,32,33]. However, as explained above, although it is dysregulated during OA and RA, the currently available data do not provide a conclusive overview about the role of SOX11 in arthritis. Whether it is involved in arthritis progression or needed for joint maintenance and the proper functioning of cartilage and bone cells is not yet clear. On the other hand, SOX12 appears to have no role in arthritis disease nor in skeletal system development [34]. However, SOX4 has been suggested by several recent studies as a therapeutic target of OA and RA [4,8,11,17,27]. Therefore, targeting SOX4 and other signaling molecules could improve the current therapeutic strategies in arthritic disorders.

## 6. Upstream Molecules That Can Target SOX4 to Treat Arthritis

Through direct and indirect binding to SOX4 TF, several upstream factors and signaling molecules have been recently reported to target SOX4 during arthritis, such as ROS)/TGF, TNF, SMOC2, AGEs, Lnc PART1, and the miRNA, miR-31 (Figure 1). The proinflammatory cytokines can target SOX4 and stabilize it to enhance the transformation of FLS in arthritic disease [10,23]. The transformed FLS promote joint degeneration in the autoimmune forms of arthritis through the production of more inflammatory cytokines and catabolic enzymes [25,35]. During this process an axis between TNF and SOX4 protein has been suggested, where SOX4 works as pivotal mediator and target of these cytokines to enable arthritis progression. The signaling downstream of the TNF, NF-κB, is one of the important signaling pathways that stimulates FLS transformation and enhances the subsequent synovial hyperplasia, degeneration of cartilage, and loss of bone during arthritis [26]. Moreover, the canonical NF-κB signaling TF, RELA/p65, and SOX4 were found to be transcriptional partners regulating transcriptional activity and inflammation in the FLS [10]. Cellular senescence is a critical process during OA pathogenesis, whereby senescent cells are arrested and then exert a secretory phenotype, SASP. SOX4-regulating FLS during OA was found to be under the control of the ROS/TGF-β signal, whereby SOX4 activation enhances cell senescence and the SASP of OA-FLS [5].

On the other hand, the endogenous secreted modular calcium-binding protein 2 (SMOC2) has been recently recognized as a key molecule to control the aggressive behavior in FLSs during arthritis. Mechanistically, SMOC2 exerts this function via SOX4-mediating transcriptional regulation leading to increased levels of upregulated motor protein myosin1c (MYO1C) and hence causing cytoskeleton remodeling and enhancing synovial migration and invasion during RA [28].

In human chondrocytes, the inflammation and degradation of the cartilage-specific proteoglycan core protein and the onset and development of OA was reported to be regulated by advanced glycation end products (AGEs). Mechanistically, in a dose-dependent manner, AGE-induced chondrocyte injury was found to upregulate the levels of both SOX4 and phosphorylated p38. In the meantime, treatment with the p38 inhibitor reduced AGE-induced SOX4 expression, indicating that the upregulation of SOX4 that resulted from AGE treatment is mediated by p38 [6]. However, two miRNAs were found to target SOX4, miR-31-5p and miR-373-3p. Lower levels of miR-31-5p and higher levels of SOX4 were seen in OA chondrocytes models and in OA patients. Mechanistic experiments indicated that miR-31-5p negatively regulates SOX4 expression by direct interaction with its 3’-untranslated region. In the meantime, the upregulation of miR-31-5p suppressed mTORC1 in an ERK-dependent way via the inhibition of SOX4 [9]. Similarly, miR-373-3p was downregulated in OA chondrocytes and cartilage tissues in a mechanism involving an upstream effect of the long non coding PART1, which promoted OA progression via regulating the miR-373-3p/SOX4 axis [4].

Besides the above-mentioned signaling molecules, preclinical studies have suggested natural anti-inflammatory and antioxidant biomolecules to target SOX4 and SOX4 signaling molecules and treat arthritis. In this regard, the classic nonsteroidal anti-inflammatory drug Feprazone was reported to have a promising effective role in treating arthritis. This has been proven mechanistically by the ability of Feprazone to reduce the degraded aggrecan in the human chondrocytes induced in TNF-α treated models, where it inhibited the loss of aggrecan via the suppression of the SOX-4/ADAMTS-5 signaling pathway [6]. In addition, we have indicated that the natural flavonoid biomolecule pinocembrin can virtually interact with SOX4 and reduces its level expression in RA mouse models leading to a reduction in arthritis symptoms [11].

## 7. Transcriptional Activity of SOX4 in Arthritis

Based on recent data, SOX4 was found to autoregulate its own expression besides regulating its family-member SOX11. As recently interpretated by Jones and colleagues [10], the autoregulation or regulation of own expression is a common mechanism seen in several developmentally important TFs to protect their abundance and activity from being repressed by other factors [36]. Regulating the expression of SOX11 by SOX4 was reported to occur through binding to a regulatory region in the 3′ UTR, which suggests that SOX4 is a crucial factor in the regulation of inflammatory responses [10]. Related to this, the single nucleotide polymorphism (SNP) in the 3′ UTR was found to be responsible for susceptibility to another common skeletal system disease osteoporosis, suggesting that the 3′ UTR region is involved in regulating *SOX4* gene expression during inflammatory diseases [37]. SOX4 polymorphisms were also shown to contribute to variations in low mineral mass density in humans [38]. In this regard, prospective mechanistic studies are needed to explore the function of SOX4 during OA and RA in relation to its structural analysis. On the other hand, information about the epigenetic regulation of SOX4 during arthritis is not yet available.

## 8. SOX4 and Its Implications in Osteoporosis

Osteoporosis, a metabolic bone disease affecting millions of the world’s population, is characterized by a lower bone mineral density, bone fragility, and a loss of bone mass. It commonly occurs during the postmenopausal period in women and in the elderly population due to the increase of bone remodeling and the imbalance between bone osteoclastic and osteoblastic activity. An impaired osteoblast associated with decreased bone formation in SOX4 +/− mice suggested an important role for SOX4 in the formation and the resorption of bone [39]. Similarly, in osteoblast progenitor cells the SOX4 level was higher, while SOX4 deficiency in osteoblasts reduced the proliferation rate of progenitor cells and delayed the differentiation of osteoblasts [40]. Moreover, based on a systematic PubMed database search, SOX4 has been suggested to be among a set of five genes which should be further validated for their predictive, diagnostic, and clinical value in osteoporosis [41]. On the other hand, investigating the differentially expressed genes (DEG) in a cohort of postmenopausal women revealed that SNPs in the MMP9 and SOX4 genes were associated with an increased risk of osteoporosis [42]. In line with this, in a study exploring the correlation between the *SOX4* gene 3′ (UTR) SNP and osteoporosis susceptibility, three SNPs loci, rs79958549, rs139085828, and rs201335371, at the *SOX4* gene were found to be significantly associated with the risk of osteoporosis [37].

## 9. SOX4 Involvement in Other Autoimmune Disorders

As a downstream target of TNF-α [43], SOX4 was found to play a crucial role in regulating the immune response. In line with this, a direct role for SOX4 was shown in upregulating the expression of the CD39 enzyme in the tTreg cells of human peripheral blood. In addition, upregulating SOX4 in Treg significantly increased the CD39 level, while SOX4 knockout Treg showed the opposing effect [44]. Interestingly, a high level of CD39 in CD4+ T cells was seen in the synovial fluid isolated from patients with juvenile arthritis [45]. Thus, SOX4 regulating CD39 in RA is an open subject to be investigated.

On the other hand, primary Sjögren’s syndrome (pSS) is an autoimmune condition in which a pathogenic type of T helper cells (CCR9+) is upregulated in patients and contributes to the immunopathology of pSS [46,47]. In synchronization with its upregulation of inflammatory diseases, the transcriptomic analysis of circulating CCR9+ cells showed higher levels of SOX4 in CCR9+ cells when compared with the other Th cells [48]. In addition, SOX4 was suggested to promote the production of high IFN-γ by CCR9+ Th cells, confirming the role of SOX4 in enhancing autoimmune disease progression, since aberrant IFN-γ expression is one of the factors that is associated with autoimmune inflammatory diseases [42]. The exposure of oligodendrocyte lineage to a higher level of SOX4 was found to prevent myelination [48], and thus it is probably involved in multiple sclerosis (MS) progression. Multiple sclerosis is also a chronic inflammatory disease targeting the central nervous system during which the axons of nerve cells are demyelinated leading to neuronal loss. In this direction, SOX4 has been found to be among the neurodegenerative/neuroinflammatory proteins that are responsible for cognitive impairment and that negatively influence remyelination and neuronal repair.

## 10. Conclusions

Although osteoarthritis and rheumatoid arthritis are two forms of arthritis occurring for different reasons, both appear to be regulated by similar signaling mechanisms. Both are chronic disorders; while OA is a degenerative disease, RA is an inflammatory one. Based on the currently available data, SOX12 seems not to be involved in arthritis disease progression, while SOX11 is dysregulated and enhances arthritic progression according to some studies, but is required for joint maintenance and protects cartilage and bone cells according to other studies. However, SOX4 appears to be a potential diagnostic biomarker and therapeutic target of arthritis. Subsequently, further studies are required to fully understand the mechanistic actions of SOXC proteins during arthritis progression and many knowledge gaps still need to be filled, which may help to find effective treatments for osteoarthritis and rheumatoid arthritis.

## Figures and Tables

**Figure 1 ijms-24-04215-f001:**
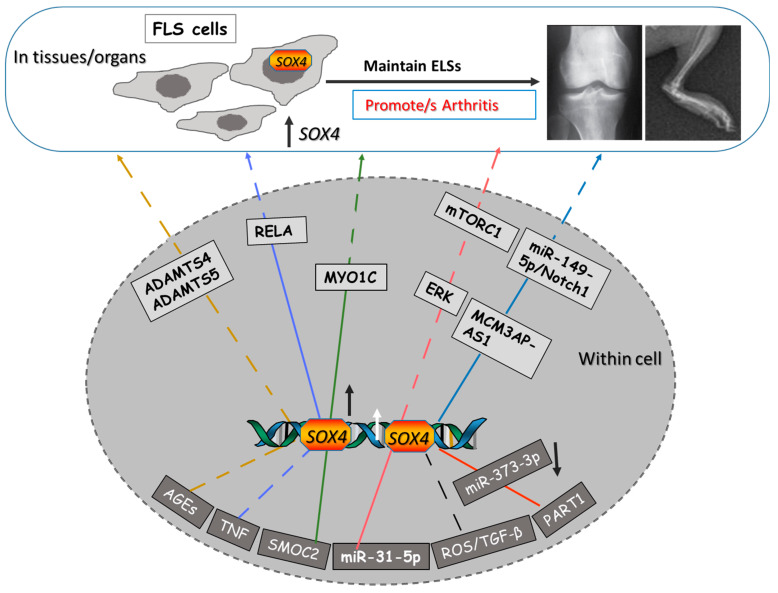
Representative diagram of the involved upstream and downstream molecules in SOX4 regulating arthritis progression. SOX4 upstream molecules within the grey background; indirect targeting, dashed lines; direct binding, continuous lines. The same signaling pathways are shown by the same colors. The upstream signaling molecules shown are advanced glycation end products (AGEs), tumor necrosis factor alpha (TNF), secreted modular calcium-binding protein 2 (SMOC2), tumor necrosis factor (TNF-β), long non coding PART1 (Lnc-PART1), and the miRNAs miR-31-5p and miR-373-3p. However, the indicated downstream target molecules include disintegrin-like metallopeptidase with thrombospondin type 4 and 5 motifs (ADAMTS4 and 5, respectively), the transcription factor RELA, the motor protein myosin1c (MYO1C), the protein kinase ERK, the mechanical target of rapamycin kinase (mTORC1), Lnc MCM3AP-AS1, and miR149-5p.

**Table 1 ijms-24-04215-t001:** SOX4 function during OA and RA.

I. SOX4 Function on OA	Experimental Model	Involved Downstream/Upstream Molecules	Ref.
SOX4 as a therapeutic target of OA	Chondrocytes cells and cartilage samples from patients and controls	PART1 mediates OA progression through regulating the miR-373-3p/SOX4 axis	[4]
SOX4 as an early diagnostic marker of OA and a novel therapeutic target	FLS cells	ROS/TGF-β-associated OA-FLS senescence	[5]
Through inhibiting SOX4, azilsartan prevents AGEs-induced degradation of aggrecan	Injured chondrocytes cell model	Upregulating SOX4 by AGEs is mediated by p38	[6]
SOX4-activated lncRNA MCM3AP-AS1 aggravates OA progression	Cartilage tissues from the knee joint of OA patients and patients who underwent amputation but did not have OA	Activated lncRNA MCM3AP modulating miR-149-5p/Notch1 signaling	[7]
SOX4 is involved in OA cartilage deterioration, through the upregulation of ADAMTS4 and ADAMTS5	Articular cell lines, mouse femoral head cartilage, and patient samples	SOX4 upregulates ADAMTS4 and Adamts5 via binding to the gene promoters	[8]
miR-31-5p promotes the survival and autophagy of OA chondrocytes	OA patients’ cartilage tissues	miR-31-5p inhibits the activation of mTORC1 in an ERK-dependent manner by the direct targeting and suppression of SOX4	[9]
SOX4 protein is a target of a proinflammatory cytokine-initiated molecular axis enhancing arthritic lesions in synovial joints	Mice, FLS culture, and human synovium specimens from OA patients undergoing total knee arthroplasty	TNF and other pro-inflammatory cytokines	[23]
**II. SOX4 Function on RA**			
SOX4 and RELA regulate the gene expression of *TNF* downstream signaling molecules leading to the FLS transformation of RA progression	FLS from wild type and mutant SOXC mice	NF-κB, RelA/p65, SOX4 and TNF	[10]
PCB reduces SOX4 which could then be a therapeutic drug in treatment of RA	adjuvant-induced arthritis mouse model	NF-κB, miR-132, miR-202-5p, and miR-7235	[11]
exFoxp3 cells are transformed into TH17 cells that induce higher levels of SOX4 to promote inflammation in joints	Patients’ splenic, synovial tissues and blood cells, and mice	N.A.	[27]
SOX4 is upregulated in synovial CD4+ T cells and further correlates with ELS formation in the RA synovium of patients	Cultured T cell	SOX4 contributes to CXCL13 production and ELS formation at inflammatory sites in humans	[17]
As a downstream of SMCO2, SOX4 binds to MYOIC to enhance migration and invasion	human STs and FLSs cells	SMCO2 and MYOIC	[28]

**Table 2 ijms-24-04215-t002:** SOX11 function during OA.

SOX11 Function in OA	Experimental Model	Involved Downstream/Upstream Molecules	Ref.
SOX11 promotes osteoarthritis through the induction of TNF-α.	Chondrocytes CHON-001 cells and knee tissues from patients	IL-1β, TNF-α	[12]
SOX11 induces chondrogenesis and cartilage defect repair by regulating β-catenin.	Rat mesenchymal stem cells	β-catenin	[29]
SOX11 potentially regulates GDF5 expression and is then involved in the pathogenesis of osteoarthritis.	OA cell model	GDF5	[13]
In OA patients, inflammatory cytokines stabilize the SOX11 protein in human inflamed synovium and FLS.	Cartilage tissues from the knee joint of OA patients or patients who underwent amputation but did not have OA	Activated lncRNA MCM3AP modulating miR-149-5p/Notch1 signaling.	[7]
Via targeting SOX4, tanshinone I inhibits IL-1β-induced inflammation and attenuates murine OA.	Chondrogenic cell line, OA murine model	IL-1β-induced collagen II, aggrecan degradation, SOX11 downregulation, and MMP-13 and p-NF-κB.	[14]
Upregulating miR-488-3p reduced the incidence of LPS-causing chondrocyte injury through inhibiting SOX11.	Knee cartilage and normal tissues	miR-488-3p and NF-κB	[15]
SOX11 promotes osteoarthritic cartilage deterioration via the induction of ADAMTS4 and ADAMTS5.	Mouse femoral head cartilage	SOX11 upregulated ADAMTS4 and Adamts5 gene promoter activities by binding to their gene promoters.	[8]
Higher *SOX11* mRNA levels in OA could be related to methylation in the 3′UTR region of the gene.	Knee articular cartilage	Differentially methylated genes	[16]
SOX11 was identified as a novel OA protein downregulated during OA.	Osteoarthritic and normal chondrocytes	N.A.	[30]

## Data Availability

Not applicable.

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
