# Peer review of "SOXC Transcription Factors as Diagnostic Biomarkers and Therapeutic Targets for Arthritis"

_ijms, 2023, doi:10.3390/ijms24044215_

Round 1
Author Response
Response to reviewer comments
Reviewer. 1.
Thank you very much for the valuable comments that for sure have helped to improve the manuscript.
- This review explored the role of SOXC transcription factors during arthritis progression and their potential utilization as diagnostic biomarkers and therapeutic targets, and demonstrated SOX4 seems to be a potential diagnostic biomarker and therapeutic target of arthritis.
- This topic is original in the field, and it summarizes that SOX4 appears to be a potential diagnostic biomarker and therapeutic target of arthritis.
- There is some minor errors: for example: osteoarthritis should be shorted to “OA” rather than “OS”;
Reply: We have now corrected that error
- where is “2.1” section?
Reply: Sections have been rearranged in the revised version, thank you
- Could the tables be revised to be more succinct and readable?
Reply: We have now revised Tables and rewritten some parts
- Could you rearranged the sections to make the relationship between them clear?
Reply: We have now rearranged sections and did the required changes
- The conclusions are consistent with the current evidences.
Reply: Thank you for the supportive comments
- The references are roughly appropriate.
Reply: It is appreciated that you find our review an interesting one
Reviewer 2 Report
Interesting paper on SOXC transcription factors in arthritis.
Abstract: Osteoarthritis (OS)-the abbreviation is OA not OS
OA is not an inflammatory disease and does not have the same pathophysiological mechanisms as RA.
The sentence in line 20-24 is too long and incomprehensible - it should be reformulated.
Introduction: Osteoarthritis (OS)-the abbreviation is OA not OS
OA and RA do not have a similar clinical phenotype and are two different diseases. Therefore, in the following text, I would separate these two diseases.
Synovitis is not the first feature of OS, while in RA inflammation is the basic pathophysiological process.
The therapeutic protocols of OS and RA are completely different and it is wrong to put them in the same context.
The initial part of the introduction (lines 32 to 50) should be reformulated
Table 1 - reformulate. The function of SOX4 on OA is highlighted in the table, and the function of SOX4 on RA should also be highlighted for better distinctness.
Progression of OA and RA disorders was recently documented in several studies to 136 be associated with upregulated levels of SOX4 (summarized in table 1). Progression has been documented for RA and OA based on existing studies. Table 1 shows 7 of 11 studies for OA, and 3 for RA. I think it is necessary to emphasize that.
Table 2 - reformulate. SOX11 Function on OA is highlighted in the table, and the function of SOX4 on RA should also be highlighted for better distinctness.
However, SOX4 was 214 suggested by several recent studies as a therapeutic target of OA and RA. The sentence about possible therapy in Ra and OA refers to references 4 and 8, which were about OA. It needs to be corrected.
SOX4 function in OA and RA related disease: I would leave out the title as well as the text below (line 297 to 300)
Conclusion: Although osteoarthritis and rheumatoid arthritis are two forms of arthritis occur due to different reasons and, both appear to be regulated by similar signaling mechanisms. The two common disorders are chronic, degenerative and inflammatory diseases. It needs to be corrected because OA is a chronic degenerative disease and RA is a chronic inflammatory disease.
Author Response
Reviewer 2.
Thank you very much for your comments and the effort you have done to improve the manuscript.
We have done what you have recommended.
Comments and Suggestions for Authors
Interesting paper on SOXC transcription factors in arthritis.
Abstract: Osteoarthritis (OS)-the abbreviation is OA not OS
Reply: We have now corrected that error
OA is not an inflammatory disease and does not have the same pathophysiological mechanisms as RA.
Reply: We have now revised that and also the related sentences in the manuscript that included that meaning.
The sentence in line 20-24 is too long and incomprehensible - it should be reformulated.
Reply: That sentence has been reformatted
Introduction: Osteoarthritis (OS)-the abbreviation is OA not OS
Reply: We have now corrected that error
OA and RA do not have a similar clinical phenotype and are two different diseases. Therefore, in the following text, I would separate these two diseases.
Reply: We have updated the manuscript and separated sections included both.
Synovitis is not the first feature of OS, while in RA inflammation is the basic pathophysiological process.
Reply: We have rewritten sentences related to this and updated the text based on the facts that you have indicated. However, when being reported at the original study, inflammatory conditions associated with OA were discussed in our review.
The therapeutic protocols of OS and RA are completely different and it is wrong to put them in the same context.
Reply: Thank you for this comment, we have now given attention to that point in separated section when possible.
The initial part of the introduction (lines 32 to 50) should be reformulated
Reply: We have reformatted this section
Table 1 - reformulate. The function of SOX4 on OA is highlighted in the table, and the function of SOX4 on RA should also be highlighted for better distinctness.
Reply: The function of SOX4 on RA has been now highlighted
Progression of OA and RA disorders was recently documented in several studies to 136 be associated with upregulated levels of SOX4 (summarized in table 1). Progression has been documented for RA and OA based on existing studies. Table 1 shows 7 of 11 studies for OA, and 3 for RA. I think it is necessary to emphasize that.
Reply: We updated that at the text and in the atble
Table 2 - reformulate. SOX11 Function on OA is highlighted in the table, and the function of SOX4 on RA should also be highlighted for better distinctness.
Reply: The function of SOX4 on OA is now highlighted but not much information is available about SOX11 in RA
However, SOX4 was 214 suggested by several recent studies as a therapeutic target of OA and RA. The sentence about possible therapy in Ra and OA refers to references 4 and 8, which were about OA. It needs to be corrected.
Reply: We have corrected that sentence
SOX4 function in OA and RA related disease: I would leave out the title as well as the text below (line 297 to 300)
Reply: We have updated this part as requested
Conclusion: Although osteoarthritis and rheumatoid arthritis are two forms of arthritis occur due to different reasons and, both appear to be regulated by similar signaling mechanisms. The two common disorders are chronic, degenerative and inflammatory diseases. It needs to be corrected because OA is a chronic degenerative disease and RA is a chronic inflammatory disease.
Reply: We have now corrected that